# Bacteriocins Revitalize Non-Effective Penicillin G to Overcome Methicillin-Resistant *Staphylococcus pseudintermedius*

**DOI:** 10.3390/antibiotics11121691

**Published:** 2022-11-24

**Authors:** Kirill V. Ovchinnikov, Christian Kranjec, Tage Thorstensen, Harald Carlsen, Dzung B. Diep

**Affiliations:** 1Faculty of Chemistry, Biotechnology and Food Science, Norwegian University of Life Sciences, 1430 Ås, Norway; 2Department of Plant Molecular Biology, Norwegian Institute of Bioeconomy Research, 1431 Ås, Norway; 3AgriBiotix AS, 1433 Ås, Norway

**Keywords:** bacteriocin, antibiotics, MRSP, skin infection, antibiotic resistance

## Abstract

The rise of antibiotic-resistant bacteria is among the biggest challenges in human and veterinary medicine. One of the major factors that contributes to resistance is use of frontline clinical antibiotics in veterinary practices. To avoid this problem, searching for antimicrobials aimed at veterinary applications is becoming especially important. Thiopeptide micrococcin P1 and leaderless peptide EntEJ97s are two different bacteriocins that are very active against many gram-positive bacteria; however, sensitive bacteria can rapidly develop resistance towards those bacteriocins. To overcome this problem, we searched for synergy between those bacteriocins and conventional antibiotics against methicillin-resistant *Staphylococcus pseudintermedius* (MRSP): a common pathogen in animal skin infections. The two bacteriocins acted synergistically with each other and with penicillin G against MRSP clinical isolates in both planktonic and biofilm assays; they also prevented resistance development. The therapeutic potential was further validated in a murine skin infection model that showed that a combination of micrococcin P1, EntEJ97s and penicillin G reduced cell-forming units of MRSP by 2-log_10_ CFU/g. Taken together, our data show that a combination of bacteriocins with conventional antibiotics can not only prevent resistance development but also pave the way to revitalize some old, less useful antibiotics, such as penicillin, which by itself has no effect on methicillin-resistant pathogens.

## 1. Introduction

*Staphylococcus pseudintermedius* is a major opportunistic pathogen in many animals, including dogs and cats, and to a lesser extent, also in humans, causing severe skin and soft tissue infections (SSTIs) and less often otitis, urinary tract infections and respiratory infections [1]. Although treatments of these infections with antibiotics such as tetracyclines, amoxicillin, cephalexin, clindamycin and fusidic acid faced relatively few challenges in the past decades [2], the recent appearance of multidrug-resistant *S. pseudintermedius* has become a considerable problem in veterinary medicine [3]. Methicillin-resistant *S. pseudintermedius* (MRSP) isolates are of special concern, as they are often resistant to many additional antimicrobial drug classes used in veterinary and human medicine [4]. In addition, MRSP can form biofilms in the wounds, effectively protecting the bacteria from the host immune response and the effects of antibiotics [5]. Therefore, new alternatives to conventional antibiotic therapies are urgently needed.

In this study, we used a mouse model to explore potential of bacteriocins to combat MRSP skin infections. Bacteriocins are ribosomally synthesized bacterial peptides produced to kill other bacteria in competition for nutrients and habitats [6,7]. Since bacteriocins kill sensitive bacteria using mechanisms different from those of antibiotics, they are equally effective against both antibiotic-resistant and sensitive bacteria, making them an appealing source of antimicrobials [8,9].

One group of bacteriocins, called thiopeptides, consists of sulfur-containing, highly posttranslationally modified molecules with a broad spectrum of antimicrobial activity, high stability and low cytotoxicity [10]. Thiopeptides inhibit sensitive cells by binding to the 50S ribosomal subunit and preventing protein translation [11]. Despite being very promising for clinical use, thiopeptides are still poorly exploited in therapeutics due to a high rate of resistance development, challenging synthesis, poor aqueous solubility and associated low bioavailability [10]. Micrococcin P1 (MP1), which was the first discovered thiopeptide [12], is a hydrophobic and heat-stable antimicrobial with high activity against a wide range of gram-positive bacteria. MP1 inhibits sensitive bacteria by attaching to a cleft between ribosomal protein L11 and helices 43 and 44 of 23S rRNA in the 50S ribosomal subunit, blocking binding of elongation factor EF-G [13]. MP1 has also been shown to be active against *Mycobacterium tuberculosis* and *Plasmodium falciparum* [12,14]. However, sensitive bacteria can easily become resistant to MP1 through single-point mutations within the gene that encodes the L11 ribosomal protein [15].

Unlike heavily posttranslationally modified thiopeptides, leaderless bacteriocins are exported from the producer strains when completely unmodified. This property makes them easy to be chemically synthesized with high purity; they can also be obtained from overexpression in bacterial hosts [16]. Leaderless bacteriocin enterocin EJ97 (EntEJ97) demonstrates potent activity against many bacteria, including pathogenic species of *Enterococcus* and *Staphylococcus* [17]. Recently, we characterized a shorter version of enterocin EJ97, which lacks seven N-terminal residues (EntEJ97s). This version showed activity against *E. faecium* and *S. haemoliticus*, similar to that of the full-length molecule, but the former is cheaper to synthesize. However, as in the case of MP1, application of EntEJ97 or EntEJ97s rapidly provokes development of resistance by mutations at the receptor protein [18,19].

Recently it has been shown that application of bacteriocins together with antibiotics (ciprofloxacin, vancomycin, rifampicin and penicillin) can revitalize the latter against antibiotic-resistant bacteria, including MRSA and vancomycin-resistant enterococci, due to strong synergetic effects [20,21,22]. Therefore, the purpose of this study was to seek possible synergy between the two bacteriocins and commercially available antibiotics, and to create a relevant MRSP skin infection model for infection and treatment studies.

## 2. Results

### 2.1. Antibiotic Resistance Profile of Staphylococcus Pseudintermedius Isolates

Sixteen *S. pseudintermedius* isolates from canine SSTIs were tested for methicillin resistance via disk diffusion assay. Six isolates were shown to be methicillin-resistant and were further tested for resistance to other antibiotics. Strain LMGT 4219 was shown to be the most resistant of the six isolates, with resistance to tetracycline, streptomycin, trimethoprim, ciprofloxacin, kanamycin, gentamicin, cloxacillin, clindamycin, erythromycin, ceftriaxone and penicillin G (PenG) (Appendix A). Due to its higher resistance, LMGT 4219 was chosen for further experiments.

### 2.2. Synergetic Antimicrobial Activities

To circumvent the bacteriocin resistance problem, we searched for antimicrobials that could act synergistically with MP1 and EntEJ7s against LMGT 4219. Seven antibiotics of different classes were tested using a checkerboard assay. As shown in Table 1, MP1 had synergistic effects with tetracycline, chloramphenicol and fusidic acid (fractional inhibitory concentration (FIC) values between 0.28 and 0.38) while the best synergy was found with PenG and EntEJ97s (FIC < 0.27). EntEJ97s, on the other hand, showed no synergy with antibiotics except for an additive effect with PenG (FIC = 0.5). Based on this data, we further explored the synergy of MP1, PenG and EntEJ97s in a three-component mixture. When the three antimicrobials were mixed, a stronger synergetic effect was observed (FIC < 0.11), with inhibiting concentrations of 5 µg/mL, 1 µg/mL and 0.2 µg/mL for PenG, EntEJ97s and MP1, respectively (Table 1).

Growth-curve analysis demonstrated that MP1 at 0.2 ug/mL was able to delay the lag phase of LMGT 4219 growth during 15–16 h of incubation, while neither PenG at 5 ug/mL nor EntEJ97s at 1.0 ug/mL individually could inhibit bacterial growth (Figure 1). On the other hand, the mixture of the three components at their respective concentrations was effective against the pathogen and did not lose its antimicrobial activity even after 48 h incubation at 37 °C (Figure 1).

To further elucidate if the three-component mixture would prevent development of resistant mutants in LMGT 4219 and in other veterinary MRSP isolates, we performed a microtiter MIC assay with a 72 h incubation time. As can be seen in Table 2, pure MP1 failed to cause resistance development in only two MRSP isolates, while PenG and EntEJ97s were not active against MRSP (MIC > 250 µg/mL). The combinations of pairwise antimicrobials exhibited significantly better antimicrobial activity than did the individuals, especially the combination of MP1 and PenG, with MIC values as low as 0.4 µg/mL for MP1 and 4 µg/mL for PenG in LMGT 4218 and LMGT 4221. In most cases, however, the combination of MP1, PenG and EntEJ97s (hereafter called the three-component mixture) showed even better antimicrobial activity compared to individual components and two-component mixtures, with MIC values as low as 0.1 µg/mL for MP1 and 1 µg/mL for PenG and EntEJ97s (Table 2). Moreover, the three-component mixture had better antimicrobial activity compared to fusidic acid—a common antimicrobial for treatment of superficial skin wounds caused by multi-resistant gram-positive bacteria in animals and humans, including MRSP (Table 2).

### 2.3. Three-Component Mixture’s Effective Inhibition of MRSP Biofilms

To examine whether our three-component mixture was effective against different MRSP isolates in biofilms, we used a modified version of the biofilm-oriented antimicrobial test (BOAT) [24]. As can be seen in Table 3, neither PenG and EntEJ97s, individually or in combination, was effective against MRSP biofilms, as expected, while MP1 (individual) and its combinations with PenG or EntEJ97s could much better inhibit MRSP cells in the biofilms. As in the previous experiment, the most effective sample was the three-component combination, where MIC of the components ranged from 5 to 40 µg/mL for PenG, 1 to 8 µg/mL for EntEJ97s and 0.2 to 1.6 µg/mL for MP1. Interestingly, these MIC values were similar or even slightly smaller compared to the MIC values for planktonic cells after 72 h of incubation. In addition, the three-component mixture had activity in the same range as fusidic acid.

### 2.4. Formulation Effectiveness against MRSP in a Murine Skin Wound Infection Model

PenG at a concentration of 5.0 mg/mL, MP1 at 0.2 mg/mL and EntEJ97s at 1 mg/mL (i.e., all 1000 times higher than the MIC values for planktonic LMGT 4219 cells) were dissolved in an APO base hand cream with 30% fat (Teva, Finland): the most suitable vehicle in terms of solubility and appropriate viscosity among tested creams with different fat concentrations (see Section 4). The cream by itself did not inhibit MRSP, while the cream containing the antimicrobials, hereafter referred to as the formulation, displayed strong antimicrobial activity (Figure 2). The formulation did not lose its antimicrobial activity when stored for 1 month at 5 °C. Given the favorable properties of the formulation, we sought to further explore its therapeutic potential in a murine infection model.

Fresh skin wounds on the dorsal sides of mice were infected with LMGT 4219 at 10^8^ CFU/wound, and the animals were left for 24 h to let the infection establish. The bacterial cell dose was found most suitable, as it presented obvious infection signs (pus and inflammation of the damaged skin). With lower cell doses, we were unable to establish a robust infection (see Appendix A). 

After infection had been established (i.e., 24 h post-inoculation of MRSP at the wound sites), the mice were divided into four groups (six mice per group), with different treatment types: the vehicle, i.e., APO base 30 cream without antimicrobials added (group 1); untreated mice as the negative control (group 2); Fucidin cream (Leo Pharma, Denmark) as the positive control (group 3); and the formulation (group 4). Fucidin cream contains 20 mg/mL of fusidic acid and is commonly used in veterinary practices to treat MRSP skin infections [25]. The mice received two treatments per day for two days (four treatments in total), and, as can be seen from Figure 3, a strong statistical difference existed between groups (one-way ANOVA, *p* = 4.9 × 10^−6^). Further pairwise comparisons indicated that a strong and statistically significant difference could indeed be observed in comparison of MRSP bacterial loads from control (APO base 30)-treated mice (5.8 × 10^7^ CFU/g) with Fucidin-(6.4 × 10^6^ CFU/g, *p* = 5.5 × 10^−4^) or formulation-treated (3.5 × 10^6^ CFU/g, *p* = 3.3 × 10^−5^) mice. Conversely, neither the difference in bacterial load between the APO base 30-treated and untreated (3.9 × 10^8^ CFU/g) groups nor that between Fucidin- and formulation-treated mice was significant (*p* = 0.22 and *p* = 0.33, respectively) (Figure 3).

## 3. Discussion

In this work, we searched for a combination of antimicrobials that had strong synergy against MRSP both in vitro and in vivo. In a collection of 16 MRSP isolates from dog skin infections in Norway, we found one with resistance not only to methicillin but also to kanamycin, streptomycin, tetracycline, cloxacillin, trimethoprim, clindamycin, ceftriaxone, ciprofloxacin, gentamicin, erythromycin and ofloxacin (Appendix A), confirming the multiresistant nature of clinical MRSP [26]. Based on the antibiotic resistance profile, it is fair to say that LMG 4219 is one of the most resistant MRSP isolates documented in Norway so far [27]. However, it showed no additional resistance to bacteriocins MP1 and EntEJ97s when compared to other, more antibiotic-sensitive *S. pseudintermedius* in this study (data not shown). This is not surprising, since bacteriocins are known to attack bacterial receptors different from those, targeted by most antibiotics [28]. 

MP1 has previously shown synergy with several antibiotics against different methicillin-resistant *S. aureus* isolates [20,24]. Interestingly, the same antibiotics—tetracyclin, rifampicin and PenG—also showed strong synergy with MP1 against MRSP (Table 1). The latter showed the best synergy and was chosen for further study. Unlike MP1, EntEJ97s showed no synergy with the panel of antibiotics, except for the additive effect with PenG. Since MP1 also showed good synergy with EntEJ97s, it was theorized that a mixture of the three antimicrobials (MP1, EntEJ97s and PenG) would likely show even better synergy against LMGT 4219 as compared to binary combinations. This was in fact confirmed in a microtiter plate assay where MIC values were as low as 0.2 µg/mL for MP1, 5 µg/mL for PenG and 1 µg/mL for EntEJ97s. The same three-component mixture had high antimicrobial activity against other MRSP isolates from dogs, and also prevented resistance development even after three days of incubation. In all tests, the mixture had at least similar antimicrobial effects as did fusidic acid—the latter being a common antibiotic to treat skin wounds in animals and humans [29] and also known to easily cause resistance development [21].

High antimicrobial activity against planktonic cells does not always correlate with high activity against biofilms [30]. Surprisingly, MIC concentration of the individual antimicrobials in the three-component mixture in planktonic cells after 72 h incubation time occupied a similar range as the MIC values of the mixture in biofilms after 24 h incubation with antimicrobials. Similar tolerance to antimicrobials in planktonic stationary-phase bacterial cells and in biofilms has been encountered before [31] and can be explained by presence of persister cells in the planktonic state, which can, to some degree, adapt to antimicrobials during long incubation time. The detailed mechanisms of such high antibiofilm activity need further investigation, but this finding clearly confirmed potential of the mixture to treat MRSP biofilms in vivo.

To our knowledge, no MRSP skin infection model in mice has hitherto been established, and since treatment efficacy was estimated by CFU counting, a superficial skin abrasion model was chosen to test the formulation [32]. Three different inoculum sizes of LMGT 4219 (10^6^, 10^7^ and 10^8^ CFU) were tested for the ability to create skin infection in mice. The highest infection inoculum (10^8^ CFU) caused notable skin inflammatory response signs 24 h post-infection: a period that has previously been shown to be sufficient for MRSP to form biofilms in artificial wound models [33]. To determine bacterial burden of the wounds, swabs could have been taken at certain time points [34], but *S. pseudintermedius* is known to significantly adhere to fibronectin, compromising reliability of the method [35]. In addition, *S. pseudintermedius* has very high internalization ability and intracellular persistence [36,37]; consequently, tissue homogenization with subsequent CFU counting was chosen in order to obtain more accurate results, though it could only be performed only once for each mouse during the experiment. Ideally, it would be best to use an MRSP strain tagged with a luciferase gene cassette (*luxCDABE*) of a bacterial bioluminescence system as a bioreporter for real-time monitoring of a bacterial wound load, as it would make testing of novel antimicrobials in vivo easier as well as reducing the number of mice in the experiment. This technology has previously been demonstrated with MRSA Xen31 (PerkinElmer): a strain containing a luciferase reporter [20,21].

In vivo results confirmed that the three-component formulation had a clear antimicrobial effect against MRSP. After four treatments were made within two days, the number of viable MRSP cells decreased by about 2 log_10_ compared to untreated mice and 1.2 log_10_ compared to vector-treated mice (APO base 30 cream). The formulation also showed therapeutic effects at about the same level as was found with Fucidin cream, although the number of mice was too small to obtain statistically significant differences between the two groups (Figure 3).

The clear synergetic effect of the three-component formulation is most likely due to different antimicrobial mechanisms of the individual components. Bacteriocin EntEJ97s belongs to the LsbB family of leaderless bacteriocins [38]. All members of this family have been shown to target sensitive bacteria using Zn-dependent membrane-bound metallopeptidase RseP (regulator of sigma E protease) as a receptor [39]. *Escherichia coli* RseP (regulator of sigma E protease) is also known as RasP (Regulator of sigma-W protease) in *Bacillus subtilis*; Eep (enhanced expression of pheromone) in *E. faecalis* and YvjB in *L. lactis* are members of the site-2 protease (S2P) family [40]. In *E. coli*, *B. subtilis* and *E. faecalis*, this protein family was shown to be involved in the stress response through activation of extracytoplasmic function (ECF) σ factors: (σE, σV, σW, etc.)-alternative sigma factors that are crucial in bacterial response to environmental stress factors [41]. Since RseP is crucial for bacterial stress response, bacteriocins that target RseP not only inhibit sensitive bacteria (Appendix A) but also make resistant mutants with mutated *rseP* more sensitive to numerous environmental stressors such as host immune response during infection) [19].

It is somewhat intriguing to see that PenG played an important role in the formation, because in general, most if not all methicillin-resistant staphylococci are known to be resistant to beta-lactam antibiotics (including PenG) through expression of a foreign penicillin-binding protein (PBP): PBP2a with low-affinity beta-lactams. Synthesis of PBP2a is regulated and normally kept at a low level [42]. The gene that encodes PBP2a–*mecA* is usually activated by beta-lactams when they appear in the environment. Induction of PBP2a is rather slow and can take up to 48 h due to tight regulation of *mecA* transcription [43], making MRSP sensitive to PenG for a short period of time. This phenomenon can also be seen with MRSP LMGT 4219, which showed sensitivity towards PenG for 8–12 h before resistance developed (Appendix A). This short time frame of activity is probably sufficient for PenG to undergo synergy with bacteriocins, especially with MP1, which inhibits protein synthesis, making bacteria even more sensitive to PenG (blocked PBP2a expression). This notion is somewhat speculative, hence further investigation is needed to reveal the nature of this synergy. Nevertheless, it is evident that application of PenG with bacteriocins that possess different modes of action can revitalize the use of this antibiotic against PenG-resistant bacteria in both veterinary and human medicine to reduce risk of bacterial-resistance development. 

Given that the number of bacteriocins with different modes of action is constantly growing, their combinations with antibiotics can become an attractive approach to combating antibiotic-resistant bacteria. Moreover, such an approach is in line with the One Health concept, as it can not only help reduce use of antibiotics but also extend lifetime of many old antibiotics.

## 4. Materials and Methods

### 4.1. Bacterial Strains and Growth Conditions

All *S. pseudintermedius* strains were grown in BrainHeart Infusion (BHI) broth (Oxoid, UK) at 37 °C overnight under aerobic conditions without shaking. *S. pseudintermedius* isolates were collected from dogs with skin pyoderma in a local veterinary clinic (Moss, Norway). *S. pseudintermedius* strains were identified using a Vitek MS v3.0 matrix-assisted laser desorption ionization–time of flight (MALDI–TOF) mass spectrometry (MS) system (bioMérieux, Marcy l’Étoile, France) according to manufacturer instructions. The strains were stored in our collection (Laboratory of Microbial Gene Technology, Norwegian University of Life Sciences, Ås, Norway).

### 4.2. Susceptibility Testing

Antibiotic susceptibility testing was performed with the disc diffusion method according to 2017 EUCAST guidelines [44]. O/N cultures were suspended in sterile saline to 0.5 McFarland and inoculated on Mueller–Hinton agar plates (BD Diagnostic Systems). Seven discs per agar plate were put in place and incubated for 24 h at 37 °C. Fourteen antibiotics were tested (all discs from Oxoid, Basingstoke, UK): fusidic acid, 5 μg; kanamycin, 5 μg; streptomycin, 10 µg; chloramphenicol, 50 µg; penicillin G, 10 µg; tetracycline, 30 µg; cloxacillin, 5 µg; trimethoprim, 5 µg; clindamycin, 10 µg; ceftriaxone, 30 µg; ciprofloxacin, 5 µg; gentamicin, 10 µg; erythromycin, 15 µg; and ofloxacin, 5 µg. Inhibition zones were evaluated based on inhibition-zone diameter, and antibiotic sensitivity was performed based on CLSI standards [45].

### 4.3. Antimicrobial Agents and Formulation Vector

EntEJ97 and EntEJ97s peptides (Figure 4) were synthesized by Pepmic Co., Ltd. (Suzhou, China) with ≥95% purity and solubilized to concentrations of 1 to 10 mg/mL in Milli-Q water. MP1 was purified to ≥95% purity, as previously described [21], and stored at a concentration of 20 mg/mL in dimethyl sulfoxide. EntEJ97s that lacked the first seven N-terminal residues showed similar potency against the 16 *S. pseudintermedius* isolates, as with the full-length EntEJ97 bacteriocin (data not shown). Since EntEJ97s was smaller and cheaper, hence more feasible to synthesize chemically, we used it further in our study. 

Antibiotics (streptomycin, gentamicin, erythromycin, chloramphenicol, kanamycin, fusidic acid, rifampicin, tetracycline and penicillin G) were obtained from Sigma-Aldrich (Norway) and solubilized to concentrations of 5 to 100 mg/mL according to the supplier’s instructions. All antimicrobials were stored at −20 °C until they were used. The final formulation for mouse treatment was prepared in APO base hand cream with 30% fat (Teva, Finland).

### 4.4. Synergy Assessment

For assessment of synergistic effects with bacteriocins, antibiotics with different modes of action were used. The selected antibiotics were gentamicin, streptomycin, kanamycin, erythromycin, chloramphenicol, tetracycline, penicillin G, fusidic acid and rifampicin. Synergy testing was carried out with a microtiter plate checkerboard assay, as previously described [21]. Fractional inhibition concentration was used to define synergy between antimicrobial A (a bacteriocin) and antimicrobial B (an antibiotic). FIC values were calculated as follows: FIC = FICa + FICb, where FICa is MIC of A in combination/MIC of A alone and FICb is MIC of B in combination/MIC of B alone. Effects were considered synergistic if FIC was ≤0.5 for a two-component mixture and ≤0.75 for a three-component mixture [46]. MIC values were determined in accordance with EUCAST recommendations (https://www.eucast.org/fileadmin/src/media/PDFs/EUCAST_files/Disk_test_documents/2022_manuals/Reading_guide_BMD_v_4.0_2022.pdf, accessed on 1 January 2022). MIC is defined as the lowest concentration of an antimicrobial or an antimicrobial combination that inhibits visible growth (at least 50% growth inhibition) of a microorganism after 24 h incubation at 37 °C in microtiter plates in 200 μL of culture.

### 4.5. Biofilm-Oriented Antimicrobial Test (BOAT)

The BOAT assay was performed as described previously [24]. The starting concentrations of the antimicrobials for all tests were 5 mg/mL for MP1, 0.1 mg/mL for PenG and 10 mg/mL for EntEJ97s. The biofilms were allowed to form for 24 h and then washed twice with 100 μL of sterile saline buffer; a total of 150 μL of antimicrobial and control dilutions was transferred from the challenge plate to the corresponding wells of the biofilm plate. The challenged biofilms were then incubated for an additional 24 h at 37 °C. After the challenge period, the antimicrobial dilutions were removed and the biofilms were carefully washed three times with 150 μL of the sterile saline buffer. A total of 100 μL of TSB supplemented with 0.025% of triphenyl-tetrazolium chloride (TTC, Sigma, Kanagawa, Japan) was then added to each well of the plate and further incubated at 37 °C for 5 h. The ensuing results were assessed through monitoring of development of red formazan (red color), denoting retainment of metabolic activity by bacterial cells. The medium was then removed, and 200 μL of ethanol:acetone (70:30) mixture was added to the wells and incubated O/N in order to extract the red formazan. The amount of extracted dye, reflecting the degree of bacterial cell metabolic activity, was then quantified by spectrophotometric readings at 492 nm. The metabolic activity inhibition for biofilm-associated cells was expressed as MIC50, which refers to the minimum concentration of the antimicrobial needed to reduce at least 50% of the metabolic activity as compared to the untreated control. The MIC50 was assessed by optical density readings at 492 nm (O.D.492) upon solubilization of red formazan.

### 4.6. Selection of Suitable Antimicrobial Vehicles

Due to its poor solubility, we performed a search for a suitable vehicle to deliver PenG at a concentration of 5.0 mg/m, MP1 at 0.2 mg/mL and EntEJ97s at 1 mg/mL. Mixture stability was tested in a panel of commercially available skin creams with different fat concentrations (22%, 30%, 47%, 60% and 70%); testing comprised dilution of the antimicrobial stock solutions into each cream. After the antimicrobials were added into the cream, the mixture (1 mL) was heated to 50 °C to reduce viscosity, mixed vigorously on a vortex for 15 s and centrifuged for 15 min at 15,000× *g* at room temperature. High solubility was reached when no visible pellet was seen at the bottom of the tubes.

### 4.7. Murine Experiment

Experiments on mice were approved by the Norwegian Food Safety Authority (Oslo, Norway), application no. 20/10793. In total, 30 female BALB/cJRj mice of 4 weeks of age were purchased from Janvier (Le Genest-Saint-Isle, France). Four mice were housed per cage (IVC; Innovive, San Diego, CA, USA) during the whole experiment and were maintained on a 12 h light/12 h dark cycle (Temp.: 25 °C +/− 1, RH 50% +/− 5), with ad libitum access to water and a regular chow diet (RM1; SDS Diet, Essex, UK). Mice were acclimatized in our mouse facilities for two weeks before the start of the experiments, hence the age of the mice at the start of the experiments being 6 weeks.

One day before infection, each mouse was anesthetized with a Zoletyl forte–Rompun–Fentadon (ZRF) cocktail (containing 3.3 mg Zoletil forte, 0.5 mg Rompun and 2.6 µg Fentanyl per 1 mL 0.9% NaCl) through intraperitoneal injection (0.1 mL ZRF/10 g body weight), then shaved on the back and flanks with an electric razor. Remaining hair was removed with hair-removal cream (Veet, Reckitt Benckiser, Slough, UK) according to the manufacturer’s instructions. The next day, the mice were again anesthetized with ZRF cocktail (0.1 mL/10 g body weight), and one skin abrasion wound was made on the back of every mouse with a sterile scalpel (Swann Morton, Sheffield, England). Prior to infection, overnight-grown LMGT4219 cells were washed twice in sterile saline and then suspended in ice-cold PBS buffer. 

To find the optimal bacterial dose for skin infection, three groups (two mice in each) were infected with 10 μL of PBS (containing ca 10^6^, 10^7^ or 10^8^ CFU of LMGT4219) via a pipette tip. After bacterial application, the mice were kept on a warm pad for 10–15 min to dry the inoculum; the wounds were then covered with 4 × 5 cm Tegaderm^TM^ film (3M Medical Products, St. Paul, MN, USA) to reduce risk of wound contamination and then left for 24 h for infection to establish. The next day, wound sites were visually inspected for infection development (inflammation, pus, etc.), and 10^8^ CFU was chosen as the infection dose, as it met the criteria described above (Appendix A).

For the treatment, wound creation and bacterial inoculation (ca 10^8^ CFU) were performed identically to that described above, and the mice (n = 24) were left for 24 h for the infection to establish. The day afterward (24 h post-infection; PI), the Tegaderm^TM^ film was removed and the mice were randomly divided into four groups (six mice in each) and subjected to four different treatments: one with formulation, one with vector (APO base 30% cream), one with Fucidin cream (2% fusidic acid in a cream base; LEO Pharma, Denmark) as a positive control and one left untreated as a negative control. All treatments were performed twice a day. Before treatment, each mouse was placed on a warm pad and anesthetized with 2% isoflurane. Approximately 200 µL of formulation, vector or Fucidin cream was spread on the inoculated skin area and gently massaged into the skin wound. After that, the mice were kept under anesthesia for 10 min to ensure suitable penetration of antimicrobials into the skin tissue. On day 3 post-infection (after four treatments), the mice were euthanized, and the affected skin area was removed via a sterile biopsy punch that measured 6 mm in diameter (Dermal Biopsy Punch, Miltex Inc, Bethpage, NY, USA), then weighed and collected in a sterile GentleMACS M tube with 2 mL saline. The skin sample was homogenized in a GentleMACS Dissociator (Miltenyi Biotec, Bergisch Gladbach, Germany) for 6 min. Each MRSP skin sample was serially diluted in saline, and 20 µL spots were applied on BHI agar plates supplemented with Ampicillin/Kanamycin (each 20 µg/mL) to select for LMGT4219. All agar plates were incubated 24 h at 37 °C before CFU counting.

### 4.8. Statistical Analysis

All in vitro assays were performed three times. For statistical analyses and graphs, R Studio (version 2022.07.01; https://rstudio.com/products/rstudio/download/, accessed on 1 June 2022) was used. All data were analyzed using ANOVA for a multiple comparison test. Where relevant, a post-hoc pairwise statistical analysis was also performed, using the unpaired two-sample *t*-test. Values where *p* < 0.05 were considered significant. 

## Figures and Tables

**Figure 1 antibiotics-11-01691-f001:**
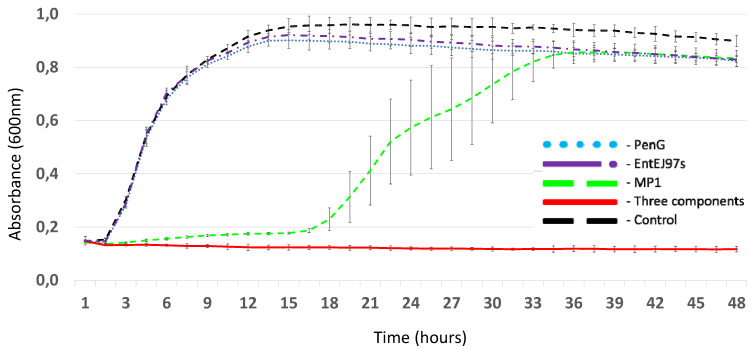
Growth curves of LMGT 4219 in the presence of different antimicrobials: PenG (5 µg/mL), EntEJ97s (1 µg/mL), MP1 (0.2 µg/mL) and a combination of all three antimicrobials at the same concentrations. Control: LMGT 4219 without antimicrobials. Error bars shown mean +/− SD of triplicates.

**Figure 2 antibiotics-11-01691-f002:**
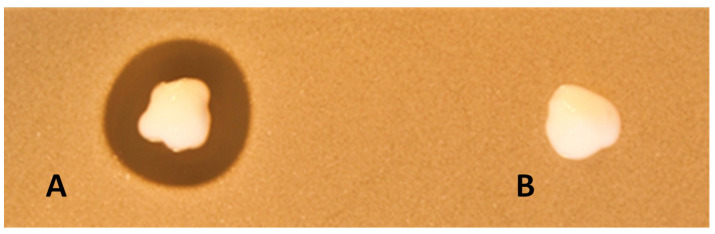
Assessment of antimicrobial activity of formulation containing MP1 (0.2 mg/mL), PenG (5 mg/mL) and EntEJs (1 mg/mL) dissolved in APO base cream (30% fat) (**A**). The APO base cream alone was used as a negative control (**B**). The activity was tested with a soft agar overlay assay using MRSP LMGT 4219 as an indicator strain.

**Figure 3 antibiotics-11-01691-f003:**
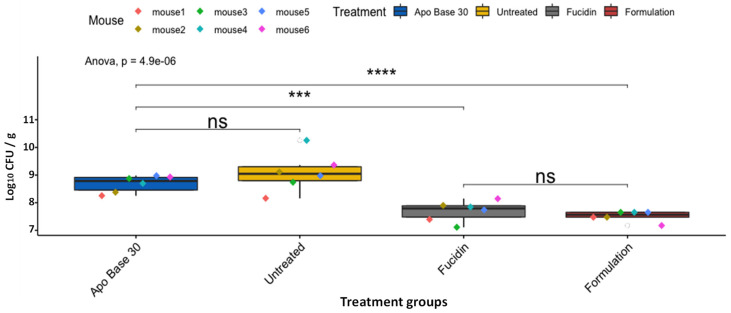
Box plots of CFU counts (CFU/g) of LMGT 4219 from skin tissue samples in differently treated mouse groups after four treatments. The area within each box represents the interquartile region (IQR), which comprises the second and third quartiles and describes the interval of values in which the middle 50% of the observed data are distributed. The horizontal black line within each box represents the median value. The extent of the IQR (box height) expresses the degree of variability measured within the middle 50% of the observed data, with whiskers extending out at either side of the boxes to mark minimum and maximum observed values as well as variability outside the middle 50% of values (whisker length). Outliers are displayed as data that extend out of the whisker limit (1.5 times the IQR). There were six mice in the untreated group. Pairwise statistical analyses were performed using the unpaired two-sample t-test. *** *p* < 0.001; **** *p* < 0.0001; ns = not significant.

**Figure 4 antibiotics-11-01691-f004:**
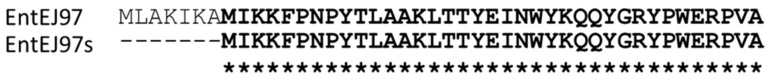
Amino acid sequence of EntEJ97 and EntEJ97s.

**Table 1 antibiotics-11-01691-t001:** Synergy assessment between MP1, EntEJ97s and a panel of antibiotics against LMGT 4219 after 24 h incubation (average of triplicate tests). Effects were considered synergistic if FIC was ≤0.5 for a two-component mixture and ≤0.75 for a three-component mixture. FIC 0.5–1 indicates additive effects [23].

Antimicrobial (A)	Individual MIC, µg/mL	MIC in Mixture (A/MP1), µg/mL	FIC	MIC in Mixture (A/EntEJ97s), µg/mL	FIC
MP1	1.6	-	-	-	-
EntEJ97s	>250	4/0.4	<0.27	>250	-
Streptomycin	>250	4/0.8	>0.5	>250/>250	>0.5
Gentamicin	>250	8/0.8	>0.5	>250/>250	>0.5
Erythromycin	>250	8/0.8	>0.5	>250/>250	>0.5
Chloramphenicol	25	6/0.2	0.38	25/25	>0.5
Kanamycin	250	16/0.4	0.38	125/125	>0.5
Fusidic acid	3	0.4/0.4	0.38	1.6/16	>0.5
Rifampicin	0.8	0.2/0.2	0.38	0.4/8	>0.5
Tetracycline	62	2/0.4	0.28	31/125	>0.5
Penicillin G	>250	8/0.4	<0.27	62/62	0.5
**Three Components**		**(A/EntEJ97s/MP1)**			
Penicillin G/EntEJ97s/MP1		5.0/1.0/0.2	<0.11		

**Table 2 antibiotics-11-01691-t002:** MIC_50_ values (µg/mL) of MP1, PenG and EntEJ97s against five MRSP strains, assessed individually and in combination after microtiter plates were incubated for 72 h at 37 °C (average of triplicate tests). Fusidic acid (F. acid) was used here for comparison.

MRSP Isolate	MP1	PenG	EntEJ97s	MP1/PenG	MP1/EntEJ97s	PenG/EntEJ97s	MP1/PenG/EntEJ97s	F. acid
LMGT 4218	3.2	>250	>250	0.4/4	0.4/4	16/16	0.1/1/1	0.4
LMGT 4220	>25	250	>250	0.8/8	1.6/16	120/120	0.8/8/8	>25
LMGT 4221	>25	>250	>250	0.4/4	1.6/16	62/62	0.4/4/4	3.2
LMGT 4222	12	>250	>250	0.8/8	0.8/8	32/32	0.4/4/4	6.4
LMGT 4223	>25	>250	>250	0.8/8	0.8/8	62/62	0.4/4/4	3.2

**Table 3 antibiotics-11-01691-t003:** MIC_50_ values (µg/mL) for MP1, PenG and EntEJs against six MRSP biofilms, assessed individually and in combination (average of triplicate tests). Challenged biofilms were incubated with antimicrobials for 24 h at 37 °C (see Section 4). Fusidic acid (F. acid) was used here for comparison.

MRSP Isolate	MP1	PenG	EntEJ97s	MP1/PenG	MP1/EntEJ97s	PenG/EntEJ97s	MP1/PenG/EntEJ97s	F. acid
LMGT 4218	0.9	>3000	>300	0.9/45	0.9/45	3000/300	0.5/25/2.5	0.5
LMGT 4219	0.9	>3000	>300	0.9/45	0.9/45	750/75	0.5/25/2.5	0.25
LMGT 4220	0.9	>3000	>300	0.9/45	0.9/45	750/75	0.5/12.5/2.5	>25
LMGT 4221	0.9	>3000	>300	0.9/45	0.9/45	750/75	0.5/25/2.5	0.1
LMGT 4222	0.9	>3000	>300	0.9/45	0.9/45	1500/150	0.5/25/2.5	0.5
LMGT 4223	4	>3000	>300	1.8/90	1.8/90	3000/300	0.5/25/2.5	0.5

## Data Availability

Not applicable.

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
