# Peer review of "Bacteriocins Revitalize Non-Effective Penicillin G to Overcome Methicillin-Resistant *Staphylococcus pseudintermedius"

_antibiotics, 2022, doi:10.3390/antibiotics11121691_

Round 1
Reviewer 1 Report
This manuscript is very interesting for readers and used a complete approach to improve the actual understanding on strategies to avoid/reduce antimicrobial resistance. This research has several merits and is well-done research. However, the current version requires several modification before meet the high-quality standard of this journal and be able to be accepted for publication.
Major comments
The title does not reflect the content of this research. Authors must inform that their study was done using only MRSP. Additionally, remove the point from in the end of the title and subtopics
References in the text must be modified to the journal style [1] [2]
M&M section must be inserted previously to Results. Additionally, several phrases in results sections must be transferred to M&M section. In the current version, the results section is a mix with M&M. The same occurs in discussion where results are presented.
Authors must include information about the identification of S. pseudointermedius from dog’s skin. How did they confirm this specie? Which tests were used?
Authors must provide more detailed information about the statistical analyses. This section is very poor, and several information are widespread in other sections.
In the discussion section, more practical applications must be discussed, and additionally, more information about antimicrobial resistance in One Health context would be interesting to be included.
There is no conclusion section.
Authors must provide their specified contributions in this research (authors contribution statement)
Minor
· The title does not reflect the content of this research. Authors must inform that their study was done using only MRSP.
· Additionally, remove the point from in the end of the title.
· To avoid shortage of this source of antibiotics, searching for anti-9 microbials aimed at veterinary applications is becoming especially important – Please, confirm the correct use of English language in this phrase.
· I suggest removing “murine model” in keywords and use other appropriate term
· Staphylococcus pseudintermedius is a major opportunistic pathogen in many animal species including dogs and cats and to lesser extent humans - – Please, confirm the correct use of English language in this phrase.
· Line 30 – don’t use etc
· Lines 71-72 – specify the commercially available antibiotics used. Restrict your aim
· Authors must reference the EUCAST instead of providing the link (lines 310 – 315)
· Reference CLSI (322)
· There is no need of Figure 5. Authors can include this amino acid sequence in the text
· Line 332 – Which antibiotics?
· Line 349 – provide the reference instead link
· A table containing all the groups formed in this research would be useful to help readers to understand the experimental design. It is not clear
· Line 354 – Modify this phrase. Continue after previously….
· Topic 4.6 – Authors must clarify the reason for doing this only for PenG
· Supplemental Figure 1 does not contribute to the understanding and can be removed
· Line 139 -140 – remove this phrase. It is not your results!
· Line 140 – 144- M&M section
· Line 197 – statistical analyses section
· Lines 215 – 216 – reference
· Figure 3 – it is a result!! Not discussion!
Reviewer 2 Report
Article
Bacteriocins revitalize non-effective penicillin G to overcome methicillin resistance
Well written article. Methodologically correct with a clear goal. The results are excellently presented. The discussion commented on the basis of other studies. References follow results
Accept
Author Response
Well written article. Methodologically correct with a clear goal. The results are excellently presented. The discussion commented on the basis of other studies. References follow results
Accept
Authors’ response: We are very thankful to the Reviewer #2 for the positive response.
Reviewer 3 Report
The manuscript describes original results regarding the synergistic effect of some antibacterial compounds (bacteriocins and antibiotics) against antibiotic resistant pathogens. Antibiotic resistance is an actual problem worldwide and finding new treatments against resistant pathogens is very challenging in human and veterinary medicine. In this context, I believe that such studies (as the ones presented in the manuscript) are important and would be of interest for the scientific world.
My only suggestion is related to Table 1 and Figure 2, for a better understanding:
-Table 1: better explain the last two lines; also mention how many times you repeated the MIC determinations.
- Figure 2: add letters on the two pictures, to differentiate the cream with/without the antibacterial mixture.
Round 2
Reviewer 1 Report
The manuscript has been improved. All my mandatory suggestions were accepted. I do not have additional suggestions to improve this manuscript.